# TWIN GRAPH CONVOLUTIONAL NETWORKS: GCN WITH DUAL GRAPH SUPPORT FOR SEMI-SUPERVISED LEARNING

## ABSTRACT

Graph Neural Networks as a combination of Graph Signal Processing and Deep Convolutional Networks shows great power in pattern recognition in non-Euclidean domains. In this paper, we propose a new method to deploy two pipelines based on the duality of a graph to improve accuracy. By exploring the primal graph and its dual graph where nodes and edges can be treated as one another, we have exploited the benefits of both vertex features and edge features. As a result, we have arrived at a framework that has great potential in both semi-supervised and unsupervised learning.

## 1 INTRODUCTION AND MOTIVATION

*Convolutional Neural Networks* (CNNs) (Lecun et al. (1998)) has been very successfully used for automated feature extraction in Euclidean domains, especially for computer vision, such as 2D image classification, object detection, etc. However, many real-life data has a non-Euclidean graph structure in nature, from which we want to investigate the underlying relations among different objects by utilizing the representation of nodes and edges. Recently, research on applying the generalization of Convolutional Neural Networks to the non-Euclidean domains has attracted growing attention. As a result, a branch of research on *Geometric Deep Learning* (Bruna et al. (2013)) based on that has been ignited. Previous works including ChebNet (Defferrard et al. (2016)) and GCN (Kipf & Welling (2017)) have demonstrated strong results in solving problems in semi-supervised learning where the labels of only a few objects are given, and we want to find out the labels of other objects through their inner connections. Current methods generalizing convolution operations include both spatial and spectral domains (Bruna et al. (2013)). The spatial one deals with each node directly in the vertex domain while the spectral one takes a further step in converting signals via *graph Fourier transform* into the spectral domain. However, one critical weakness would be the fact that the interchangeable and complementary nature between nodes and edges are generally ignored in previous research. As a result, the duality of the graph is not fully utilized. If we treat those edges in the original, or known as the primal graph, as the nodes in the new graph, and original nodes as edges, we can arrive at a new graph that further exploits the benefits of edge features. In such a way, we are able to get both the primal graph and the dual graph (Monti et al. (2018)). By combining both the vertex features and the edge features, we will be able to solve wider range of problems and achieve better performance. In this paper, we propose a new approach to transform the primal graph into its dual form and have implemented two pipelines based on these two forms of graph to improve the accuracy and the performance. With two pipelines, we also exploited a path to make the model wider instead of merely deeper. Meanwhile, we have developed a new framework that can be applied later on both semi-supervised learning and unsupervised learning.

## 2  RELATED WORK

**Graph-based semi-supervised learning** aims to annotate data from a small amount of label data on a graph. To learn the vectors that can recover the labels of the training data as well as distinguish data with different labels, conventionally, graph Laplacian regularizer gives penalty between sampling based on graph Laplacian matrix (Zhu et al. (2003); Ando & Zhang (2007); Weston et al. (2012)). Sample-based method takes random walk to get samples from the context of data points in order to propagate information (Perozzi et al. (2014); Yang et al. (2016); Grover & Leskovec (2016)).

**Graph Convolutional Networks** generalize the operation of convolution from grid data to graph data (Wu et al. (2019)). After the emergence of the spectral-based convolutional networks on graph (Bruna et al. (2013)), ChebNet (Defferrard et al. (2016)) approximate the filters by Chebyshev polynomials according to the Laplacian eigendecomposition. GCN(Kipf & Welling (2017)) simplifies ChebNet by introducing its first-order approximation and can be viewed as a spatial-based perspective, which requires vertices in the graph to propagate their information to the neighbors. MoNet(Monti et al. (2017)) is a spatial-based method, of which convolution is defined as a Gaussian mixture of the candidates. GAT(Veličković et al. (2017)) applies the attention mechanism to the graph network. DGI(Veličković et al. (2018)) proposes a framework to learn the unsupervised representations on graph-structured data by maximizing location mutual information. We refer to Zhou et al. (2018); Xu et al. (2018); Battaglia et al. (2018); Wu et al. (2019) as a more comprehensive and thorough review on graph neural networks.

**Dual approaches on graph networks** usually unlike the above mono-methods, apply mixed methods to study graph networks. DGCN(Zhuang & Ma (2018)) makes a balance between the spatial-based domain and spectral-based domain by regularizing their mutual information. GIN(Yu et al. (2018)) proposes a dual-path from graph convolution on texts and another network on images to gather cross-modal information into a common semantic space. DPGCNN(Monti et al. (2018)) extends the classification on vertices to edges by considering the attention mechanism on both. Our study follows this path, which classifies vertices from the relationship between them (edges) and regularization from the mutual information between classification on both vertices and edges.

## 3  METHODOLOGY

### 3.1  PRELIMINARIES

Let $\mathcal{G} = \{\mathcal{V}, \mathcal{E}, \boldsymbol{A}\}$ denote a *graph*, where $\mathcal{V} = \{1, \ldots, N\}$ is the set of nodes with $|\mathcal{V}| = N$, $\mathcal{E}$ is the set of edges, and $\boldsymbol{A} = (A_{(i,j) \in \mathcal{V}} \neq 0) \in \mathcal{R}^{N \times N}$ is the *adjacency matrix*. When $\mathcal{G}$ is undirected then $\boldsymbol{A}$ is symmetric with $A_{i,j} = A_{j,i}$, $\mathcal{G}$ is an undirected graph, otherwise a directed graph. The *Laplacian matrix*, also acts a propagation matrix, has the combinatorial form as $L = D - A \in \mathbb{R}^{N \times N}$, and its normalized form is $\mathcal{L} = I - D^{-\frac{1}{2}} A D^{-\frac{1}{2}}$, where $D = diag(d(1), \ldots, d(N)) \in \mathbb{R}^{N \times N}$ is the *degree matrix* of graph $\mathcal{G}$ with $d(i) = \sum_{j \in \mathcal{V}} A_{i,j}$ and $I \in \mathcal{R}^{N \times N}$ is the identity matrix. In some literature, the random walk Laplacian $L_{rw} = I - D^{-1} A$ is employed to directed graph $\mathcal{G}$.

### 3.2  GRAPH CONVOLUTIONS AND CONVOLUTIONAL NETWORKS ON GRAPHS

Let $\mathcal{L} = U \Lambda U^T$ be the eigendecomposition, where $U \in \mathbb{R}^{N \times N}$ is composed of orthonormal eigenbasis and $\Lambda = diag(\lambda_0, \lambda_1, \ldots, \lambda_{N-1})$ is a diagonal matrix of eigenvalues which denotes frequencies of graph $\mathcal{G}$, and $\lambda_i$ and $u_i$ form an eigenpair. The *convolutional operator* $*_{\mathcal{G}}$ on the graph signal $x$ is defined by

$$f = g *_{\mathcal{G}} x = U \left( (U^T g) \odot (U^T x) \right) = U \hat{G} U^T x, \tag{1}$$

where $\hat{f} = U^T x$ and $\hat{g} = U^T g$ are regarded as the **graph Fourier transform** of graph signal $x$ and graph filter $g$, respectively; $f = U(\cdot)$ is the *inverse graph Fourier transform*, and $\odot$ is the *Hadamard product*. $\hat{G} = diag(\hat{g}_0, \cdots, \hat{g}_{N-1})$ behaves as spectral filter coefficients. Graph convolution can be approximated by polynomial filters, the $k$-th order form is

$$U \hat{G} U^T x \approx \sum_{i=0}^{k} \theta_i \mathcal{L}^i x = U \left( \sum_{i=0}^{k} \theta_i \Lambda^i \right) T^T x, \tag{2}$$

where $\theta_i$ denotes coefficients and $\hat{G} \approx \sum_i \theta_i \Lambda^i$ or equivalently $\hat{g}(\lambda_j) \approx \sum_i \theta_i \lambda_j^i$. Based on the above approximation, ChebNet (Defferrard et al. (2016)) further introduces Chebyshev polynomials into graph filters of the convolutional layers for the sake of computational efficiency. Chebyshev polynomials are recursively expressed as $T_i(x) = 2x T_{i-1}(x) - T_{i-2}(x)$ with $T_0(x) = 1$ and $T_1(x) = x$. The graph filter then becomes

$$U\hat{G}U^T x = U \left( \sum_{i=0}^{k} \theta_i T_i(\Lambda) \right) U^T x = \sum_{i=0}^{k} \theta_i T_i(\tilde{\mathcal{L}}) x, \qquad (3)$$

where $\tilde{\mathcal{L}} = 2/\lambda_{max}\mathcal{L} - \mathbf{I}$ denotes the scaled normalized Laplacian for all eigenvalues $\lambda_i \in [-1, 1]$ and $\theta_i$ is trainable parameter. Graph Convolutional Network (GCN) (Kipf & Welling (2017)) is a variant of ChebNet which only takes first two terms of Equation (3). By setting the coefficients $\theta_0$ and $\theta_1$ as $\theta = \theta_0 = -\theta_1$ and with $\lambda_{max} = 2$, the convolution operator in convolution layer of GCN is induced as $g *_{\mathcal{G}} x = \theta(I + D^{-\frac{1}{2}} A D^{-\frac{1}{2}})x$. With *renormalization trick*, $I + D^{-\frac{1}{2}} A D^{-\frac{1}{2}}$ is further replaced by $\tilde{D}^{-\frac{1}{2}} \tilde{A} \tilde{D}^{-\frac{1}{2}}$ where $\tilde{A} = A + I$ and $\tilde{D} = D + I$.

### 3.3 GRAPH AND ITS DUAL

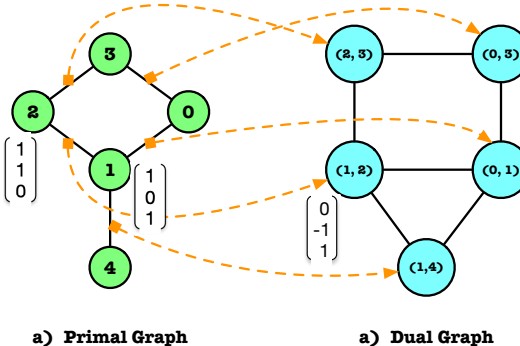

a) Primal Graph        a) Dual Graph

Figure 1: Primal graph and its dual graph: a) A graph $\mathcal{G}$, which represents a primal graph, with 5 nodes; b) the corresponding dual graph $\hat{\mathcal{G}}$ to graph $\mathcal{G}$ at its left side; arrow-arcs demonstrate the conversion between $\mathcal{G}$ and $\hat{\mathcal{G}}$.

In graph theory, The definition of the *dual* varies according to the choice of embedding of the graph $\mathcal{G}$. For planar graphs generally, there may be multiple dual graphs, depending on the choice of planar embedding of the graph. In this work, we follow the most common definition. Given a plane graph $\mathcal{G} = \{\mathcal{V}, \mathcal{E}\ A\}$, which is designated as the *primal graph*, the **dual graph** $\hat{\mathcal{G}} = \{\tilde{\mathcal{V}} = \mathcal{E}, \tilde{\mathcal{E}}, \tilde{A}\}$ is a graph that has a vertex (or node) for each edge of $\mathcal{G}$. The dual graph $\hat{\mathcal{G}}$ has an edge whenever two edges of $\mathcal{G}$ share at least one common vertex. To be clarified, the vertices $(i, j)$ and $(j, i)$ of dual graph $\hat{\mathcal{G}}$ converted from a undirected graph are regarded as the same. Fig.1 shows the conversion from primal graph to its dual counterpart. When vertices of the primal graph embed features (or *signals* in terminology of spectral graph theory), the features of a dual node can be obtained by applying a specified functions to its corresponding primal nodes' features, i.e. the simplest applicable function is to calculate the distance between the features of two nodes. In addition, if the edges of primal graph possess features or attributes, we also take them into account as the their inherited features of dual nodes. Take node $(1, 2)$ of dual graph in Fig.1b) as an example, its feature is obtained by performing the element-wise subtraction to the feature vectors of nodes 0 and 3 of primal graph in Fig.1a), i.e. $[1, 1, 0]^T - [1, 0, 1]^T = [0, -1, 1]^T$.

### 3.4 TWIN GRAPH CONVOLUTIONAL NETWORKS

The *Twin Graph Convolutional Networks* (TwinGCN) proposed in this work consists of two pipelines. Both pipelines are built with the same architecture as GCN, and contain two convolution layers in each pipeline, as shown in Fig.2. The upper pipeline acts exactly as GCN; however, the lower one takes the dual features $\hat{X}$ derived from primal features $X$ as its inputs (as described

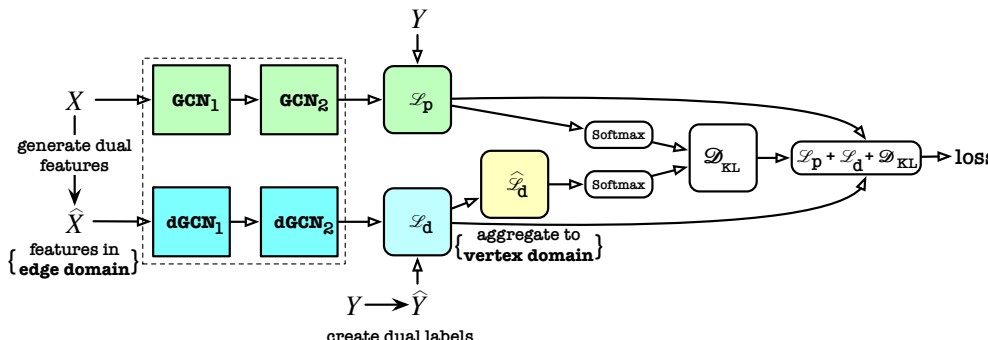

Figure 2: Architecture of TwinGCN: Two pipleline architecture, the upper is for primal graph, the lower is for the dual graph. The predictions of the two pipelines are combined with their KL-Divergence for final results.

in section 3.3), the predictions or outputs in dual vertex domain (i.e. edge domain in primal) is then aggregated to primal vertex domain. The goal of introducing a dual pipeline into the model is that we desire to utilize the predictions on the dual node (edges in primal graph) to affect the predictions on primal nodes since the knowledge about those neighbors of a node can be propagated through edges.

For the purpose of training the dual pipeline, we also need to get the labels of dual nodes. Let us take an example, given a dual node $(i, j)$ (corresponds to an edge in primal graph), primal node $i$ has label $\alpha$ and $j$ has label $\beta$, then dual node $(i, j)$ is assigned with a label $(\alpha, \beta)$.

One thing worth mentioned is that TwinGCN's convolution layers are not limited to those used in GCN, they can be replaced with other types of convolution layer, such as ChebNet, GWNN (Xu et al. (2019)), etc.

The convolution layers in the pipelines perform graph convolution operations with shared weights as learnable parameters, mathematically expressed as

$$H^{(l+1)} = \sigma \left( \tilde{D}^{-\frac{1}{2}} \tilde{A} \tilde{D}^{-\frac{1}{2}} H^{(l)} W^{(l)} \right) \tag{4}$$

where $H^{(l)}$ is the activation in $l$-th layer, $W^{(l)}$ is learnable weights in that layer. $\sigma$ represents non-linear activation function, e.g. ReLU. For the task of semi-supervised node classification, the loss function is defined as

$$\mathcal{L} = - \sum_{l \in \mathcal{Y}_L} \sum_{f=1}^{F} Y_{l,f} ln Z_{l,f} \tag{5}$$

where $\mathcal{Y}_L$ is set of node labels for $L \in V$ labeled node set, $F$ denotes the number of labels of the nodes, and $Z$ is predicted outcome, a softmax of the output of the network.

In order to take effect of dual pipeline on the prediction of primal pipeline, we adopt *KullbackLeibler Divergence* $(\mathcal{D}_{KL})$ as a regularization term in training. Suppose that $P(Y|X)$ is predictions by primal pipeline and $P(\hat{Y}|\hat{X}) = P(\hat{Y}|X)$ is the derived predictions obtained through an aggregation from the predictions on dual labels by dual pipeline to primal label predictions. $\hat{X}$ is derived from $X$ as aforementioned (Section 3.3). We first calculate the joint probability matrix $P(Y, \hat{Y})$ of two matrices $P(Y|X)$ and $P(\hat{Y}|X)$

$$P(Y, \hat{Y}) = \sum_{x \in X} P(y|x) P(\hat{y}|x) = P(Y|X)^T P(\hat{Y}|X) \tag{6}$$

we further get the marginal probabilities of $P(Y)$ and $P(\hat{Y})$ from $P(Y, \hat{Y})$. KullbackLeibler Divergence $\mathcal{D}_{KL}$ is evaluated by

$$\mathcal{D}_{KL} \left( Y || \hat{Y} \right) = - \sum_{y \in Y} P(y) log \left( \frac{P(\hat{y})}{P(y)} \right) = - \sum_{y \in Y} P(y) \Big[ log P(\hat{y}) - log P(y) \Big] \tag{7}$$

finally, we attains the loss function as

$$\mathcal{L} = \theta_1 \mathcal{L}_P + \theta_2 \mathcal{L}_D + \theta_3 \mathcal{D}_{KL}, \qquad (8)$$

where $\theta_1$, $\theta_2$, and $\theta_3$ are trainable coefficients.

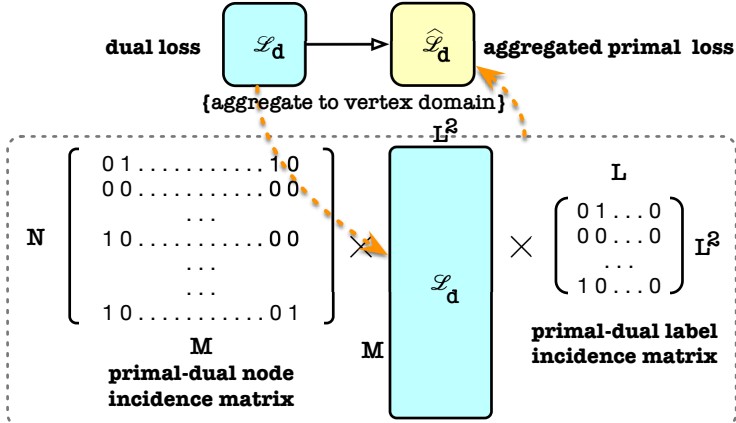

Figure 3: Fast algorithm for calculating the aggregated primal predictions from dual predictions

Fig.3 illustrates a fast algorithm deriving primal predictions from predictions of dual pipeline. It is conducted by introducing two special incidence matrices. The matrix at the left hand side ($N \times M$, $N = |\mathcal{V}|$ and $M = |\mathcal{E}|$) is an incidence matrix in which the rows represent primal nodes, each column depicts whether a primal node in a row has an incidence in the dual node represented by this column. The rightmost matrix is the incidence matrix of primal labels presenting in dual labels with dimension of $L^2 \times L$. Although these two matrices are extremely sparse when node number is very large (we store them in compressed form), by taking advantage of GPU's powerful computing capability, the delicate *sparse matrix multiplication* subroutine, e.g. Nvidia's cuSARSE, runs much faster than codes with loops for lumping the incidences.

## 4 EXPERIMENTS

In this section, we evaluate the performance of TwinGCN, we mainly focus on semi-supervised node classification in current work. Actually, TwinGCN also support unsupervised learning by changing the loss functions which we will fulfill in future work.

### 4.1 DATASETS

We conduct experiments on three benchmark datasets and follow existing studies (Defferrard et al. (2016); Kipf & Welling (2017); Xu et al. (2019) etc.) The datasets include Cora, Citeseer, and Pubmed (Sen et al. (2008)). All these three datasets are collected from their corresponding citation networks, the nodes represent documents and edges are the citations. Table 4.1 shows details of these datasets. Label rate indicates the portion of the available labeled nodes used for training. The training process takes 20 labeled samples for each class for every dataset.

| Dataset | Nodes | Edges | Features | Label Rate | Classes | Train/Valid/Test Nodes |
|---------|-------|-------|----------|-----------|---------|------------------------|
| Cora | 2,708 | 5,429 | 1,433 | 0.052 | 7 | 140 / 500 / 1,000 |
| Citeseer | 3,327 | 4,732 | 3,703 | 0.036 | 6 | 120 / 500 / 1,000 |
| Pubmed | 19,717 | 44,338 | 500 | 0.003 | 3 | 60 / 500 / 1,000 |

Table 1: The Statistics of Datasets

### 4.2 BASELINES AND EXPERIMENT SETTINGS

Since both pipelines of our proposed architecture work with graph convolution based on spectral graph theory, we use recent works, such as ChebNet (Defferrard et al. (2016)) GCN (Kipf & Welling

(2017)), and GWNN (Xu et al. (2019)), etc. These models maintain the same graph Lapalacian base structure, unlike some other methods take partial graph structure, e.g. FastGCN (Chen et al. (2018)) applies Monte Carlo importance sampling on edges. however, this kind of method only guarantees the convergence as the sample size goes to infinity.

For the sake of consistency for comparison, the hyper-parameters for training are kept the same for primal pipeline as other models. The primal are composed with two graph convolution layers with 16 hidden units and applied with ReLU non-linear activations. Loss is evaluated with the softmax function. Dropout (Srivastava et al. (2014)) of primal pipeline is set to $p = 0.5$ for the primal. We use the Adam optimizer (Kingma & Ba (2015)) for optimizing the weights with an initial learning rate $lr = 0.01$.

As the dual graph is normally much bigger than the counterpart primal graph, its adjacency/Laplacian matrix and the number of dual nodes becomes quadratically larger, e.g. $N$ nodes with $N \times (N-1)$ edges in a fully-connected graph. Therefore, to avoid overfitting on dual pipeline, we set its dropout rate higher than 70%. We also introduce a sampling rate to extract a small fraction from the total dual node labels. Having a large number of edges in the primal graph also means a large number of dual nodes. In such situation, the performance will be degraded severely.

### 4.3 PERFORMANCE OF TWINGCN

The quantitative comparison among different models is given in Table 4.4. For node classification, TwinGCN achieves similar results or outperforms with some datasets. The performance gain comes from the aggregation of knowledge propagated through edges (or dual nodes) trained by dual pipeline. However, primal pipeline only will ignore the dependency between labels of nodes.

| Method | Cora | Citeseer | Pubmed |
|---|---|---|---|
| ChebNet | 81.2% | 69.8% | 74.4% |
| GCN | 81.5% | 70.3% | 79.0% |
| GWNN | 82.8% | 71.7% | 79.1% |
| DGI | 82.3% | 71.8% | 76.8% |
| DGCN | 83.5% | 72.6% | 80.0% |
| **TwinGCN** | 82.7% | 72.8% | 79.8% |

Table 2: Results of Node Classification

Fig.4a) illustrate that when compared to the GCN, TwinGCN bearing two pipelines converges slower but achieves a higher accuracy as the number of epoch increases. This is because that we have two pipelines through mutual interaction. In Fig.4b), we observe that two loss curves of traditional GCN and TwinGCN have very similar decreasing trends. However, the loss curve of TwinGCN is slightly above GCN because the loss of TwinGCN is the summation of both primal and dual pipelines.

### 4.4 ANALYSIS

To test whether the introduced dual pipeline and regularization improve the basic GCN pipeline, we conducted controlled experiments to make comparison among GCN, GCNs with pipelines on original graph and dual graph and TwinGCN(GCNs with both pipelines and regularization by KL-divergence).

| Method | Cora | Citeseer | Pubmed |
|---|---|---|---|
| GCN | $81.5\% \pm 0.3\%$ | $70.8\% \pm 0.1\%$ | $78.8\% \pm 0.1\%$ |
| GCN(double-pipeline) | $81.6\% \pm 0.4\%$ | $72.5\% \pm 1.0\%$ | $79.8\% \pm 2.3\%$ |
| **TwinGCN** | $83.0\% \pm 1.3\%$ | $72.5 \pm 0.8\%$ | $79.5\% \pm 1.2\%$ |

Table 3: Comparison results

Table 3 shows the comparison results: the average test accuracy and the standard deviation from the model which has the best validation accuracy. The pipeline on the dual graph increases the

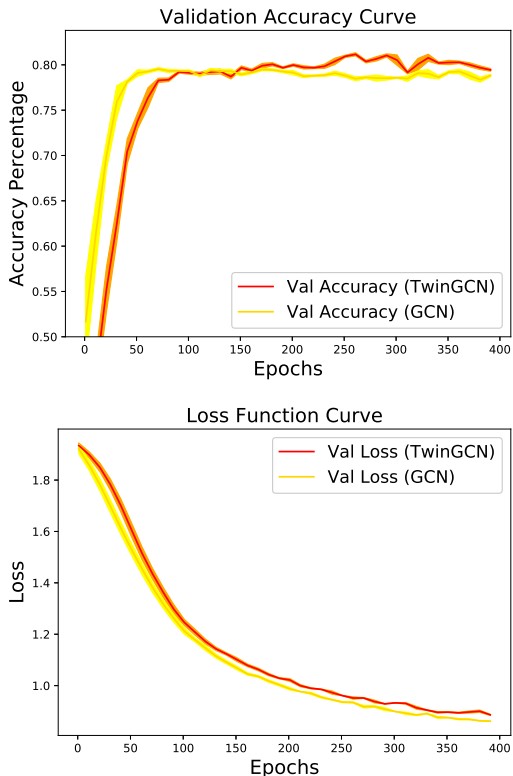

Figure 4: a) Mean and standard deviation of validation accuracy on Cora; b) Loss function of validation

performance of GCN, indicating that applying relationship between nodes can be a powerful tool in the classification task. The regularization(KL-divergence). However, TwinGCN suffers from larger uncertainty suggested by the larger standard deviation.

## 5 CONCLUSION AND FUTURE WORK

In this work, we propose the TwinGCN with parallel pipelines working on both the primal graph and its dual graph, respectively. TwinGCN achieves the state-of-the-art performance in semi-supervised learning tasks. Moreover, TwinGCN's ability is not limited to this, we can extend its power/utilization into unsupervised learning by altering its loss functions.

## 6 ACKNOWLEDGMENTS

Use unnumbered third level headings for the acknowledgments. All acknowledgments, including those to funding agencies, go at the end of the paper.

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

# A APPENDIX

## A.1 SPASITY

TwinGCN, which introduces additional pipeline, increases the number of parameters as well as sparsity. For the graph $G$ with $m$ nodes and $n$ edges, the sparsity of the Laplacian matrix is $\mathcal{O}(\frac{n}{m^2})$. If we denote $n_i$ as the degree of node $i$, for the dual graph which takes edges as nodes and connect edges which share the same node, the sparsity of the dual Laplacian matrix is

$$Sparsity(\hat{\mathcal{G}}) = \mathcal{O}\left(\frac{\sum_{i=1}^{m} n_i(n_i - 1)/2}{n^2}\right)$$

which can be as small as $\mathcal{O}(\frac{1}{m})$ when each node has the average number of degree, i.e. $n_i = \frac{2n}{m}$.

