# OpenReview forum: "TWIN GRAPH CONVOLUTIONAL NETWORKS: GCN WITH DUAL GRAPH SUPPORT FOR SEMI-SUPERVISED LEARNING"
_ICLR.cc/2020/Conference — Reject_

### Official Review · AnonReviewer2 · 2019-10-23
**Official Blind Review #2**

**Rating:** 3

**Review:**

This paper proposes two graph convolutional network for semi-supervised node classification. The model is composed of two GCNs, one on the primal graph and one on the dual graph. The paper is well written and easy to follow. However, the novelty and contribution are rather limited, and the performance improvement of the proposed method is rather limited. Following are the detailed comments:

1. The novelty and contribution of the paper are rather limited. The idea of dual graph has been exploited by [1]. The proposed method is simple combination of prime and dual without convincing explanation on why primal dual graph can improve the performance. The authors may need to give more explanations on why such combination can improve the performance.

2. The performance improvement of the proposed method is marginal. In fact, the proposed method doesn’t outperform DGCN

3. Experiments need to be improved. The authors didn’t compare with GNNs that also adopts primal and dual graphs. The authors should consider comparing the proposed method with DPGCN in [1].

[1] Monti, Federico, et al. "Dual-primal graph convolutional networks." arXiv preprint arXiv:1806.00770 (2018).


**Experience Assessment:**

I have published one or two papers in this area.

**Review Assessment: Checking Correctness Of Derivations And Theory:**

I carefully checked the derivations and theory.

**Review Assessment: Checking Correctness Of Experiments:**

I assessed the sensibility of the experiments.

**Review Assessment: Thoroughness In Paper Reading:**

I read the paper thoroughly.

---

### Official Review · AnonReviewer1 · 2019-10-26
**Official Blind Review #1**

**Rating:** 1

**Review:**

The paper describes a new dual method for graph convolutional networks that combines the features from the graph and it's dual, in two pipelines. The paper builds on the architecture as in GCN and in addition to the dual pipelines, one from the graph and other it's dual, employs KL divergence to achieve the final prediction.

The paper leaves inadequate explanation on the results, where the proposed TwinGCN comes short in 2 of 3 methods compared to other methods in Table 1, which cannot be ignored considering the slow convergence and marginal improvements, compared to GCN with double pipeline in Table 3. This leaves the premise of the authors on adding improvements to the learning ability by bringing in features from dual graph on shaky grounds.

**Experience Assessment:**

I have read many papers in this area.

**Review Assessment: Checking Correctness Of Derivations And Theory:**

I did not assess the derivations or theory.

**Review Assessment: Checking Correctness Of Experiments:**

I carefully checked the experiments.

**Review Assessment: Thoroughness In Paper Reading:**

I read the paper thoroughly.

---

### Official Review · AnonReviewer4 · 2019-11-01
**Official Blind Review #4**

**Rating:** 3

**Review:**

This paper utilizes two pipelines based on the duality of a graph and the primal graph for semi-supervise learning. More specifically, the authors transform the primal graph into its dual form and build a two-pipelines architecture to learn on the primal graph and dual graph together. The goal of including a dual pipeline is to use the predictions on the dual node to affect the predictions of the primal nodes.

I decided to give a weak reject to this paper for the following shortcomings:

1. Novelty is not enough. Dual graphs are explored utilized to graph neural networks in some related works (such as [1]). This paper utilizes dual graphs to affect or assist the prediction on the primal nodes, which is very similar to the methods used in [1].

2. Experiments can not prove the effectiveness of this method very well. The experiments only conduct on 3 datasets and the analysis of the experimental results is not convincing and more details should be included. For example, if GCN(double-pipeline) performs better than GCN, this can somehow support the effectiveness of a double-pipeline that includes a dual graph. However, the results of TwinGCN can not support the effectiveness of regularization by KL-divergence. I think more experiments and analyses about why this method works should be explored in Section Experiments.

3. The organization should be improved. In general, when introducing a model, it is better to introduce the prediction part of the model first. For example, I can get the idea of how to train the DualGCN by reading the Section Method. However, I have no idea about how the authors make a prediction on the trained DualGCN, how to transform the prediction of the dual graph to the prediction of primal nodes. I think these details are significant for this paper. Also, there are several incomplete sentences in this paper.

Overall, this idea that utilizes the dual graph to assist the primal graph learning is good and can be explored in the future. For the current version, I give a weak reject for the above reasons.




Reference
[1] Chen et al. Supervised community detection with line graph neural networks.


**Experience Assessment:**

I have read many papers in this area.

**Review Assessment: Checking Correctness Of Derivations And Theory:**

I assessed the sensibility of the derivations and theory.

**Review Assessment: Checking Correctness Of Experiments:**

I carefully checked the experiments.

**Review Assessment: Thoroughness In Paper Reading:**

I read the paper thoroughly.

---

### Decision · Program_Chairs · 2019-12-19

**Decision:**

Reject

**Comment:**

All three reviewers are consistently negative on this paper. Thus a reject is recommended.